# Effect of $Y_2O_3$ Content on Microstructure and Corrosion Properties of Laser Cladding Ni-Based/WC Composite Coated on 316L Substrate

**Feilong Liang** [1,2]🔅, **Kaiyue Li** [3], **Wenqing Shi** [2,4,*]🔅 **and Zhikai Zhu** [3]

1   Naval Architecture and Shipping College, Guangdong Ocean University, Zhanjiang 524088, China; 13542018087@163.com
2   Guangdong Provincial Key Laboratory of Intelligent Equipment for South China Sea Marine Ranching, Guangdong Ocean University, Zhanjiang 524088, China
3   School of Electronics and Information Engineering, Guangdong Ocean University, Zhanjiang 524088, China; likaiyue0512@163.com (K.L.); 13873571918@163.com (Z.Z.)
4   School of Materials Science and Engineering, Guangdong Ocean University, Yangjiang 529500, China
*   Correspondence: swqafj@126.com; Tel.: +86-13724783416

**Abstract:** To improve the corrosion resistance of 316L substrate and lengthen its useful life in marine environments, Ni-based/WC/$Y_2O_3$ cladding layers with different $Y_2O_3$ contents were fabricated on 316L stainless steel using laser cladding technology. The influence of $Y_2O_3$ additives on the microstructure and properties of the cladding coatings was investigated by using scanning electron microscopy, energy dispersive spectroscopy, X-ray diffraction, a microhardness tester, an electrochemical workstation and a tribometer. Results show that the metallurgical bonding is well formed between the coating and the 316L substrate. The coating consisted primarily of γ-Ni phase and carbides. Adding an appropriate amount of $Y_2O_3$ can effectively refine the microstructure and inhibit the precipitation of the carbide hard phase; in addition, the added rare earth element can promote the solid-solution-strengthening effect of the cladding coatings, thus improving the microhardness and wear resistance of the cladding coatings and their electrochemical corrosion property in 3.5 wt% NaCl solution. The hardness of the Ni-based/WC coatings was substantially higher than that of the substrate, and it was greatest at a $Y_2O_3$ content of 1%. The corrosion and wear resistance of $Y_2O_3$-modified Ni-based/WC composite coatings are significantly better than those of the composite coating without $Y_2O_3$.

**Keywords:** laser cladding; 316L stainless steel; Ni-based composite coating; rare earth; electrochemical corrosion

## 1. Introduction

Owing to its good corrosion resistance and easy processability, 316L stainless steel is one of the most widely used stainless steels in the parts of marine engineering equipment [1–3]. Since the generated oxide film has extremely strong adhesion to iron and steel, 316L stainless steel material can meet the requirement for purity of media in the equipment; however, the outside surface of 316L stainless steel material may still be damaged due to difficulty in withstanding the wear and corrosion caused by complex marine environments [4–7]. In particular, metal materials in the seawater splash zone are affected most seriously by corrosion due to the following reasons: the wave and spring tide movements of seawater cause the metal surface to exist in alternating dry and wet states, the wave impact effect and the friction and collision effects of ice cakes in winter continuously destroy the protective layer and corrosion product layer on the metal surface, and so on [8]. Long-time corrosion may cause great degradation of the mechanical properties of the material, causing serious hidden safety issues in the parts of marine engineering equipment. To solve the problem of the failure of the parts of marine engineering equipment caused by wear and corrosion

and to improve the seawater corrosion resistance and mechanical properties of the parts and lengthen their service life, a corrosion-resistant alloy coating can be fabricated on the surface of the material [9,10]. The traditional corrosion-resistant alloy coating fabricating technology based on thermal spraying has such disadvantages as large heat-affected zone, many defects in the coating, and insufficient bonding strength between coating and substrate [11,12]. The laser cladding technology has such advantages as small heat-affected zone, no generation of pollutants, low cost, and high processing precision [13,14]; using this technology to improve such properties of the surface of substrate materials as corrosion resistance and wear resistance has been a particular focus of domestic and international scholars in recent years.

Although nickel-based alloys, as commonly used cladding materials, have better wear resistance and corrosion resistance than 316L stainless steel and S355 steel, which are commonly used in offshore engineering, there are still cases where their performance cannot be applied to certain harsh working environments [15,16]. However, using WC with excellent wear resistance, corrosion resistance, high temperature resistance, and an antioxidant property as a reinforcing phase of Ni-based alloys can effectively improve various properties of the composite coating on metal. At present, many scholars have studied the effects of different contents of WC added in Ni-based alloy powder on the properties of cladding coatings, and all results show that the microhardness and wear resistance of the cladding coatings are improved with the increase in mass fraction of WC in cladding powder [17–19]. However, the change in corrosion resistance exhibits a different characteristic; the added WC causes the microstructure of the cladding coatings to become dense and even, effectively improving the corrosion resistance of the cladding coatings, and the continuous increase in mass fraction of WC increases the types and quantity of ceramic phases precipitating in the cladding coatings, thereby increasing the quantity of primary cells in the cladding coatings and the size of the ceramic phases, to raise the corrosion rate, thus degrading the electrochemical corrosion property of the cladding coatings [20,21]. Owing to their functions such as refining microstructures and inhibiting segregation, rare earth elements have been applied in the laser surface modification field in recent years. An appropriate amount of rare earth element added can dramatically improve the wear resistance and corrosion resistance of cladding coatings, while an excess amount of rare earth element added may result in such problems as an increase in the dilution rate, and a decrease in the reinforcing phases [22,23], degrading the properties of the cladding coatings. Thus, the key to the improvement of the properties of a cladding coating by the addition of a rare earth element is to determine an optimal addition amount.

Wang et al. [24] fabricated Ni60 alloy coatings with different contents of $CeO_2$, $La_2O_3$ and $Y_2O_3$ on the surface of aluminum alloy by cladding. When the rare earth content is less than 4%, the defects of the cladding coatings decrease with the increase in rare earth content. When the rare earth content is more than 5%, the defects of the cladding coatings increase with the increase in rare earth content. The Ni60 cladding coating with 4% $CeO_2$, 5% $La_2O_3$ and 5% $Y_2O_3$ added has the smoothest micromorphology, without evident pores and cracks. Yao et al. [25] added different contents of $Y_2O_3$ into Ni60 alloy coatings and the results show that $Y_2O_3$ can improve the structure refinement and element segregation of the cladding layer, and the highest microhardness and the best wear resistance were obtained when the addition amount of $Y_2O_3$ was 1%. Shu et al. [26] fabricated Ni-based WC composite coatings with different mass fractions of $CeO_2$ using laser in situ generation technology and investigated the effect of the addition amount of $CeO_2$ on the shape and size of WC particles. Results show that the size of WC particles exhibits a trend of decreasing first and then increasing with the increase in addition amount of $CeO_2$. When the mass fraction of $CeO_2$ is 2%, the size of the WC particles is the smallest and the wear resistance of the cladding coating is the best.

To fabricate $Y_2O_3$-modified Ni-based WC composite coatings with good properties, Ni-based WC composite coatings with different $Y_2O_3$ contents were fabricated on 316L stainless steel using laser cladding technology in this experiment. They were then com-

pared with Ni-based WC composite coatings without $Y_2O_3$ added, and the changes in micromorphology, microhardness and electrochemical corrosion property in 3.5 wt% NaCl solution were investigated.

## 2. Materials and Methods

### 2.1. Experimental Materials

In this experiment, 316L stainless steel was selected as substrate material, and the sample was 100 mm $\times$ 50 mm $\times$ 3 mm in size, with the chemical composition shown in Table 1. The surface of the substrate material was ground with 240, 800 and 1500 mesh sandpapers in turn, first to remove the oxide film and impurities on the surface, and then flushed with absolute ethanol to remove the film debris and oil stain on the surface. Afterwards, the substrate material was placed in a ventilated area for air drying.

**Table 1.** Chemical composition of 316L stainless steel (mass fraction, %).

| Element | C | Mn | P | S | Si | Cr | Ni | Mo | N | Fe |
|---|---|---|---|---|---|---|---|---|---|---|
| Value | ≤0.03 | ≤2.0 | ≤0.045 | ≤0.03 | ≤0.75 | 16–18 | 10–14 | 2–3 | ≤0.1 | Bal |

The cladding coating materials selected were 300 through 500 mesh Ni60 and WC and 500 mesh $Y_2O_3$ powder. They were mixed at a given ratio, placed into a ball mill for even mixing, and then oven-dried. The chemical composition of Ni60 powder and the powder-mixing ratio are shown in Tables 2 and 3, respectively. Figure 1 shows the SEM morphology of the composite powder in group B.

**Table 2.** Chemical composition of Ni60 powder (mass fraction, %).

| Element | C | Si | Fe | B | Cr | Ni |
|---|---|---|---|---|---|---|
| Value | 0.8 | 4 | 15 | 3.5 | 15.5 | Bal |

**Table 3.** Ni60, WC and $Y_2O_3$ component ratio (mass fraction, %).

| Sample | A | B | C |
|---|---|---|---|
| Ni60 | 85 | 84.15 | 83.3 |
| WC | 15 | 14.85 | 14.7 |
| $Y_2O_3$ | 0 | 1 | 2 |

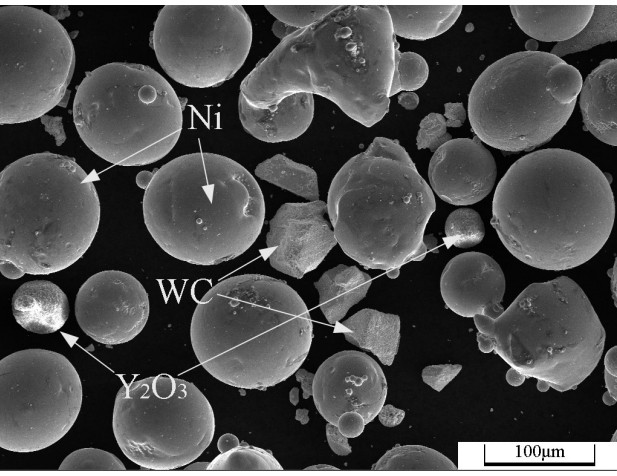

**Figure 1.** SEM morphology of the composite powder.

### 2.2. Experimental Methods

In the experiment, laser cladding was conducted with an XL-F2000W optical fiber laser processing system, the powder feeding method used was the powder presetting

method, and the powder laying thickness was $(1 \pm 0.1)$ mm. Three groups of test specimens underwent multi-track cladding processing separately, and the cladding process parameters selected in the experiment were as follows: the scanning speed was 800 mm/min, the laser processing power was 1200 W, the laser spot diameter was 3 mm and the track spacing was 1.2 mm. After the test specimens cooled down fully, two 10 mm×10 mm samples were cut out from each test specimen and were then inlaid with the cross-section and surface of cladding coating exposed separately. Afterwards, the inlaid samples were ground and polished with a metallographic sample polishing and grinding machine. The treated cross-section samples were corroded. Afterwards, the microstructure of the cladding coatings was observed with a scanning electron microscope (SEM, Tescan Mira4, Tescan, Brno, Czech Republic) and the element composition was analyzed with an EDS (Xplore 30). The phase composition of the cladding coatings was analyzed with an X-ray diffractometer (XRD, XRD-6100, Shimadzu, Kyoto, Japan) under the following conditions: the scanning mode was θ/2θ scanning, the ray source was CuKα ray source, the scanning angle range was 10–90°, the scanning step was 0.04° and the scanning speed was 7°/min.

The microhardness of the cladding coatings was tested with a digital microhardness tester (MHVD-1000AT, Shanghai Jvjing Precision Instrument Manufacturing Co., Ltd., Shanghai, China). The microhardness test was conducted from the top of the cladding coating to the substrate. Three points at a spacing of 50 μm were selected in the horizontal direction, and the microhardness values at the three points were averaged. The test point spacing in the vertical direction was 0.15 mm. The applied load was 200 g and the load holding time was 10 s. The electrochemical corrosion property of the samples was tested with a three-electrode measurement system based on an electrochemical workstation (GHI660, Chen Hua Instruments, Shanghai, China). The selected corrosion solution was NaCl solution with a mass fraction of 0.035, the working electrode, reference electrode and auxiliary electrode were specimen under test, saturated calomel electrode (mercury/calomel in saturated KCl) and platinum plate in turn, and the sample corrosion area was 10 mm × 10 mm. Before the test, the test specimen was placed into corrosion solution to make its self-corrosion potential stable. The test method adopted was the potentiodynamic test method, the scanning range was −1.5–0.5 V, the scanning speed was 0.01 V/s, the working temperature was room temperature (25 °C), the test time was 200 s, and the test frequency of electrochemical impedance spectra was $1.0 \times (10^{-1}–10^{5})$ Hz. The frictional coefficient of the samples was tested with a tribometer (MFT-5000, Rtec-Instruments, Oakland, CA, USA) under the following conditions: the frictional pairs were 9.8 mm GCr15 balls with composition shown in Table 4, the sampling frequency was 100 Hz, the load was 20 N, the reciprocating speed was 10 mm/s, the test temperature was room temperature and the test duration was 30 min.

**Table 4.** Chemical composition of GCr15 steel ball (mass fraction, %).

| Element | C | Mn | P | S | Si | Cr | Ni | Mo | Cu | Fe |
|---|---|---|---|---|---|---|---|---|---|---|
| Value | 0.95–1.05 | 0.25–0.45 | ≤0.025 | ≤0.025 | 0.15–0.35 | 1.4–1.65 | ≤0.3 | ≤0.1 | ≤0.25 | Bal |

## 3. Results and Discussion

### 3.1. Analysis of Micromorphology of Cladding Coatings

Figure 2 shows the SEM micromorphology of samples. As can be seen from Figure 2, WC-reinforced Ni-based coating can achieve good metallurgical bonding with 316L stainless steel substrate material, and an appropriate amount of $Y_2O_3$ added can make the bottom of the cladding coating become flatter and the size of unmolten WC particles in the cladding coating become relatively small. Figure 3 shows the SEM micromorphology of the bottom of cladding coatings. As can be seen in Figure 3, there are thick and large dendritic crystals growing in the direction perpendicular to the bonding interface in the bottom of the cladding coating without $Y_2O_3$ added (as shown in Figure 3a), the size of dendritic crystals decreases greatly in the bottom of the cladding coating with 1% $Y_2O_3$ added (as

shown in Figure 3b), and the increase in $Y_2O_3$ content on the basis of 1% will increase the size of dendritic crystals (as shown in Figure 3c). The reasons are possibly as follows: the added rare earth can improve the fluidity of the molten pool and reduce the surface tension between various components of the molten pool, thus improving the wettability between solid and liquid at the interface; the stable compound formed by the rare earth element and other element in the solidification process increases the number of nucleation particles, thus accelerating the nucleation; and the rare earth element may undergo segregation at crystal boundaries, thus reducing the Gibbs free energy of the system and inhibiting the growth of the crystals, thereby lowering the growth speed of the dendritic crystals [27]. The relationship of the number of grains per unit volume ($Z_V$) with the nucleation rate (N) and growth speed (G) is as follows:

$$Z_V = 0.9(N/G)^{0.75} \tag{1}$$

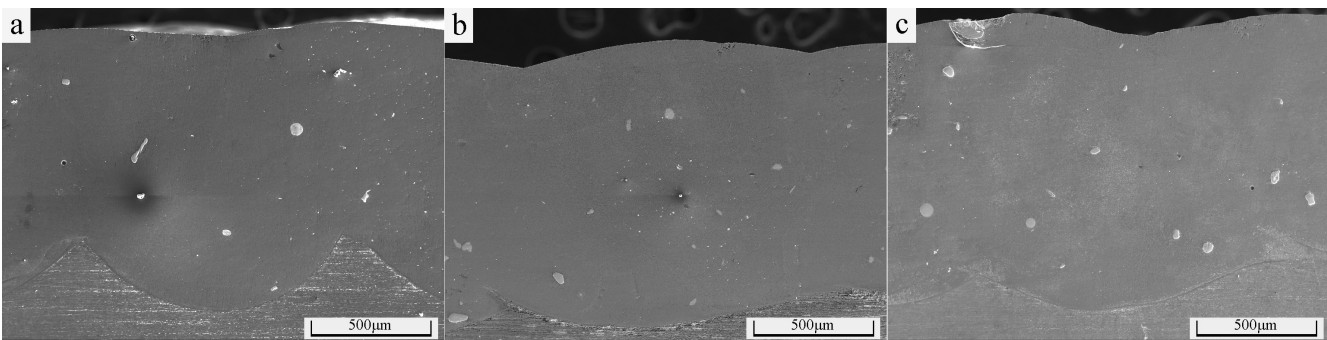

**Figure 2.** SEM micrographs of cladding coatings: (**a**) 0% $Y_2O_3$; (**b**) 1% $Y_2O_3$; (**c**) 2% $Y_2O_3$.

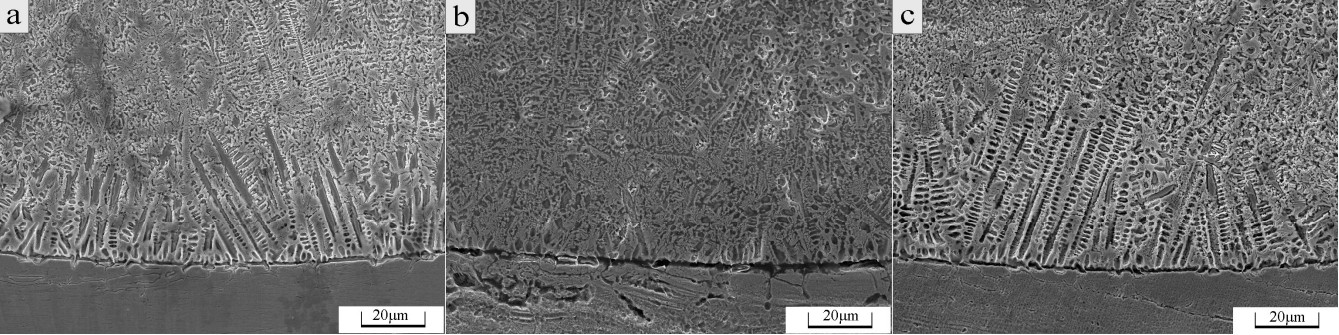

**Figure 3.** SEM micrographs of the bottom of cladding coatings: (**a**) 0% $Y_2O_3$; (**b**) 1% $Y_2O_3$; (**c**) 2% $Y_2O_3$.

Therefore, the added rare earth element causes the nucleation rate to increase and the growth speed to decrease, so that the grains are refined. Excess $Y_2O_3$ added may form a large amount of refractory compounds with impurity elements, resulting in reduced fluidity of liquid alloy in the molten pool and decreased convection speed of liquid metal in the molten pool, and too many rare earth elements may cause the crystal boundaries to be contaminated, losing their capability to inhibit the growth of grains [28].

The SEM micromorphology of the top of cladding coatings and that of the middle of cladding coatings are shown in Figures 4 and 5, respectively. There is a large amount of carbide in band and block shapes precipitating in all of the cladding coatings. Based on the EDS image of the middle of cladding coating with 1% $Y_2O_3$ added (Figure 6), the WC powder decomposes under laser to produce elements W and C, and these elements, together with such alloy elements as Cr and Fe, form a new phase in the cooling process, which precipitates out [29]. The added $Y_2O_3$ causes the banded precipitate to become evenly distributed, with size decreasing somewhat. In the middle of the cladding coating

with a $Y_2O_3$ addition amount of 1% (as shown in Figure 5d), the quantity of banded microstructures decreases and there are many dendritic crystals precipitating. Figure 7 shows the SEM micromorphology and element surface distribution at the particle site of the cladding coating without $Y_2O_3$ added. Figure 8 shows the SEM micromorphology and element surface distribution at the particle site of the cladding coating with a $Y_2O_3$ mass fraction of 0.01. Comparison between Figures 7 and 8 shows that rare earth element Y is enriched at the site of incompletely molten WC particle, reducing the precipitating carbide around the particle, indicating that an appropriate amount of rare earth added can inhibit the segregation of carbide. Figure 9 shows the element distribution from the top of the cladding coating to the substrate. The elements Ni, Cr and Fe are distributed even more evenly in the middle and top of cladding coating than in the bottom, and the content of element Fe increases gradually from the bottom of the cladding coating to the substrate, while that of element Ni decreases gradually, indicating that part of element Fe in the substrate diffuses into the cladding coating.

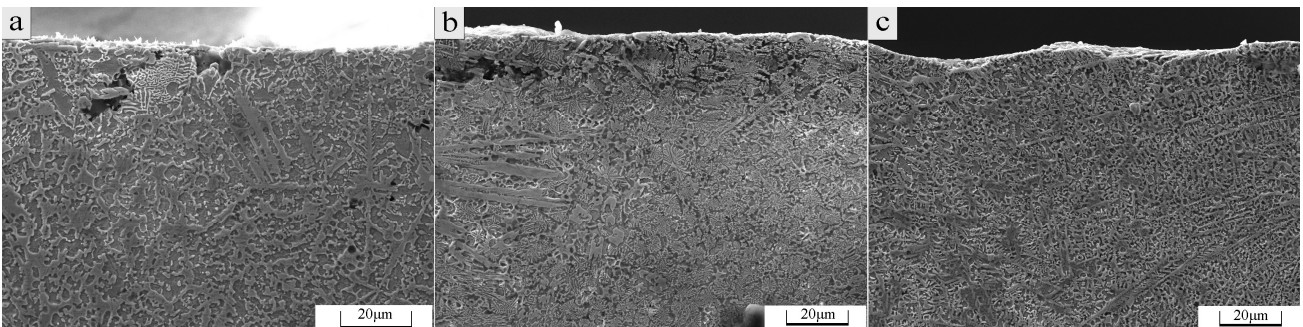

**Figure 4.** SEM micrographs of the top of cladding coatings: (**a**) 0% $Y_2O_3$; (**b**) 1% $Y_2O_3$; (**c**) 2% $Y_2O_3$.

### 3.2. Phase Analysis of Cladding Coatings

Figure 10 is the XRD pattern of cladding coatings. There are the following phase structures of cladding coating: $\gamma$-Ni, $Cr_7C_3$, WC and $Cr_{23}C_6$, etc. Element Y combines with element Ni, forming a phase $Ni_5Y$. Since the $Y_2O_3$ addition amount is small, no relevant phase other than $Ni_5Y$ is detected. The diffraction peak corresponding to $\gamma$-Ni in the figure exhibits a leftward shifting phenomenon relative to all of three diffraction peaks of pure metal Ni, and the shifting amount of the main peak increases from $0.51°$ to $0.52°$ with the increase in $Y_2O_3$ content in the cladding coating. According to the Bragg formula:

$$2d\sin\theta = n\lambda \tag{2}$$

The peak shifts to the low $2\theta$ angles, indicating that the interplanar spacing d increases. The cause of peak shifting may be that, in the laser cladding process, elements produced from WC decomposition such as W, Cr and Fe, which have larger atom radii than element Ni, enter $\gamma$-Ni in a solid solution manner to cause lattice distortion and increase the interplanar spacing d, thus having a solid-solution-strengthening effect. The quenching characteristic of laser cladding also helps increase the limit solid solution concentration of a sosoloid, thus further improving the solid-solution-strengthening effect [30]. Since the atom radius of element Y is greater than that of element Ni, a certain amount of Y can also enter $\gamma$-Ni in a solid solution manner to increase the interplanar spacing, resulting in further leftward shifting of peak.

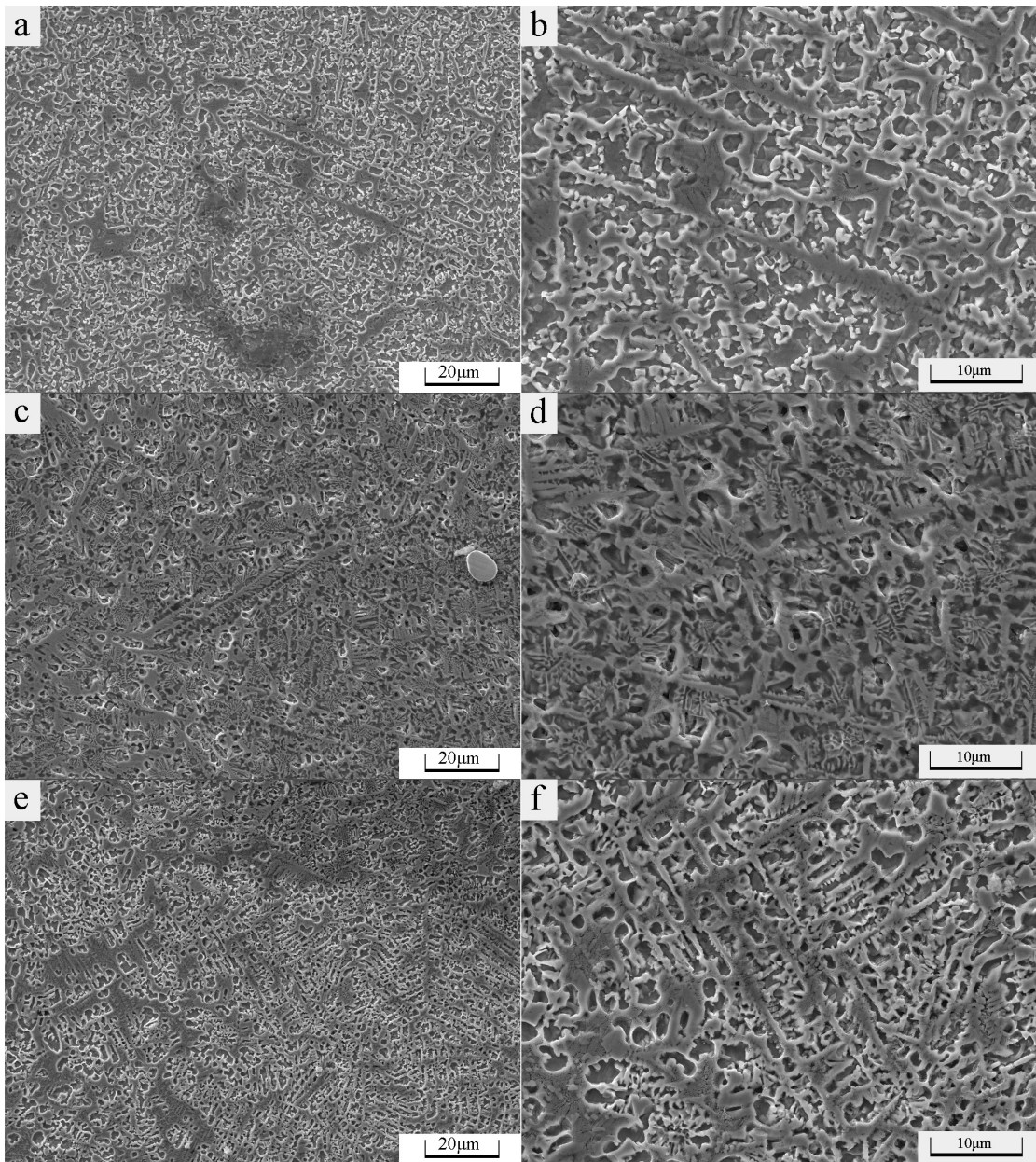

**Figure 5.** SEM micrographs of the middle of cladding coatings: (**a**,**b**) 0% $Y_2O_3$; (**c**,**d**) 1% $Y_2O_3$; (**e**,**f**) 2% $Y_2O_3$.

### 3.3. Electrochemical Corrosion Analysis of Cladding Coatings

Figure 13 shows the potentiodynamic polarization curves of the cladding coatings in 3.5 wt% NaCl solution. Table 5 shows the electrochemical parameters corresponding to the potentiodynamic polarization curves in Figure 13. In the table, $E_{corr}$ is the free corrosion potential, which is a thermodynamic parameter used to characterize the tendency of a corrosion reaction [34]. The higher the free corrosion potential, the more difficult it will be for the metal to lose electrons. The free corrosion potential does not necessarily have a connection with the corrosion reaction rate. $I_{corr}$ is the corrosion current density, and it generally exhibits a positive correlation with the corrosion rate [35]. As can be seen in Figure 13, the cladding coatings with different $Y_2O_3$ contents have extremely similar potentiodynamic polarization curves, which are in the range of −0.7−−0.1 V, and each of which has a process of sharp ascending and then constantly tending to stable, indicating that the cladding coatings underwent passivation after corrosion [36]. This is mainly due to

the fact that the element Cr in each cladding coating forms a passive film in the surface part of the cladding coating, and the Ni content in the cladding coating is relatively high, so the Ni enriched under the passive film can prevent the reduction by oxidants in the passive film position [37,38]. It is inferred according to the electrochemical parameters shown in Table 5 that the corrosion rate decreases with the increase in $Y_2O_3$ addition amount. When the $Y_2O_3$ mass fraction is 1%, the corrosion rate is the lowest, meaning that it is most difficult for corrosion to occur. A continuous increase in the $Y_2O_3$ content over 1% may result in a tendency for a corrosion reaction and in an increase in the corrosion rate.

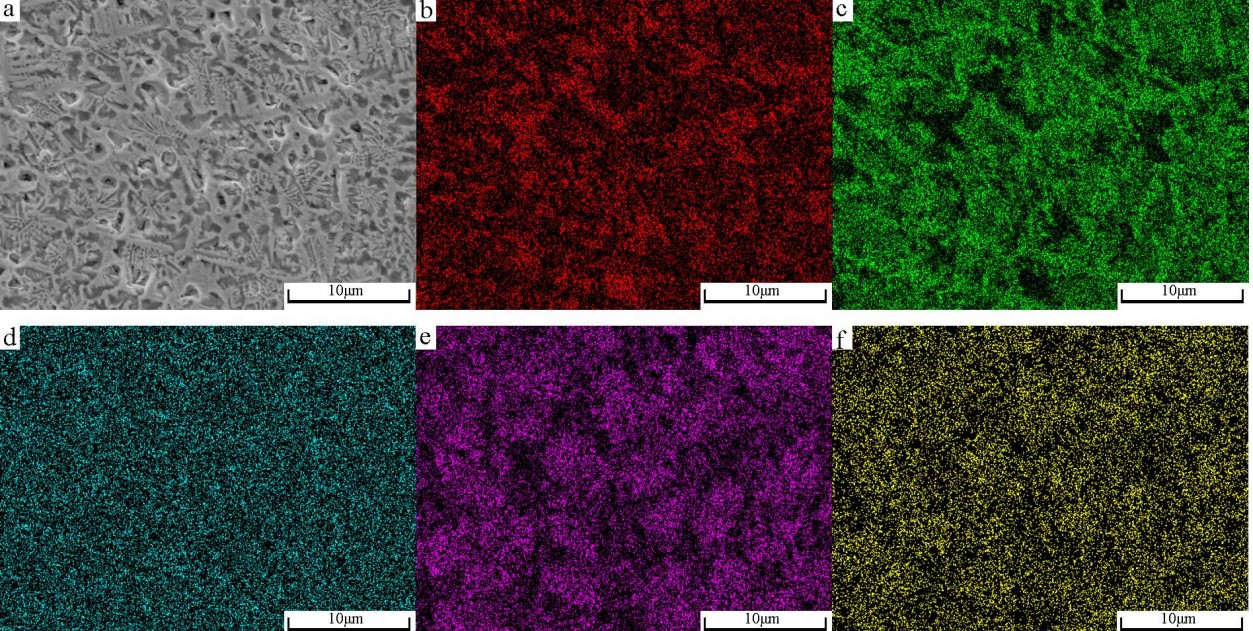

**Figure 6.** SEM micrograph and EDS mapping of cladding coating with $Y_2O_3$ mass fraction of 0.01: (**a**) SEM micrograph; (**b**) Ni; (**c**) Cr; (**d**) Fe; (**e**) W; (**f**)Y.

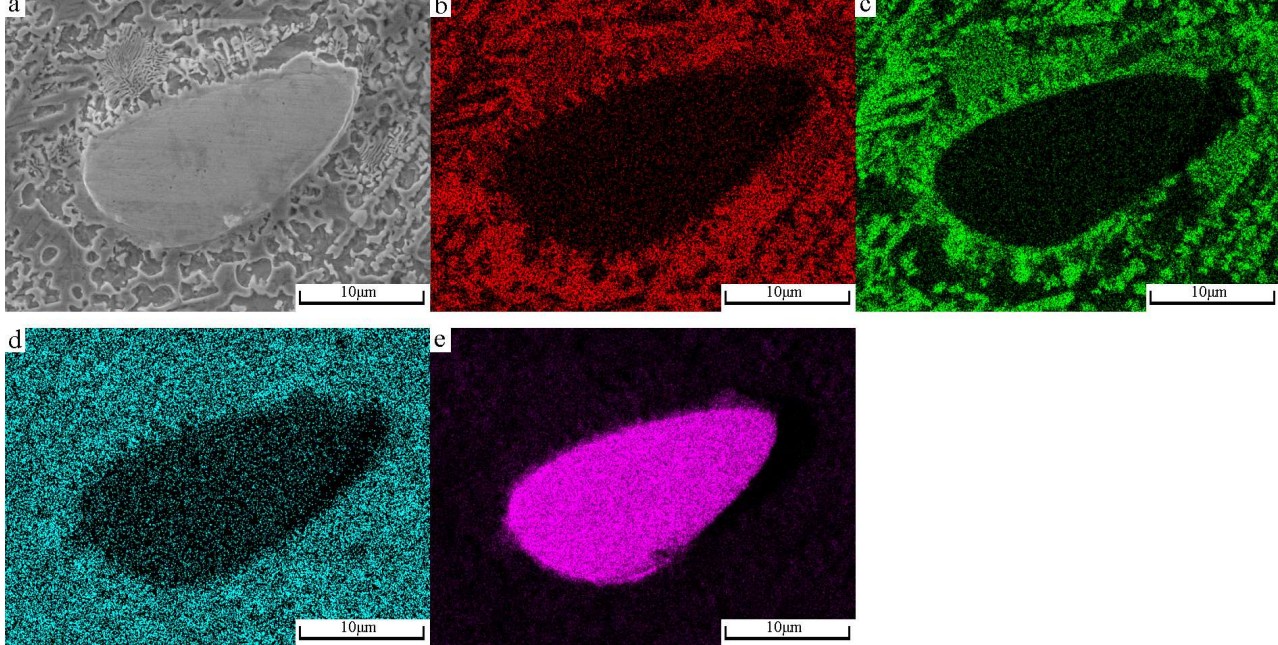

**Figure 7.** SEM micrograph and EDS mapping of the particle site of cladding coating without $Y_2O_3$ added: (**a**) SEM micrograph; (**b**) Ni; (**c**) Cr; (**d**) Fe; (**e**) W.

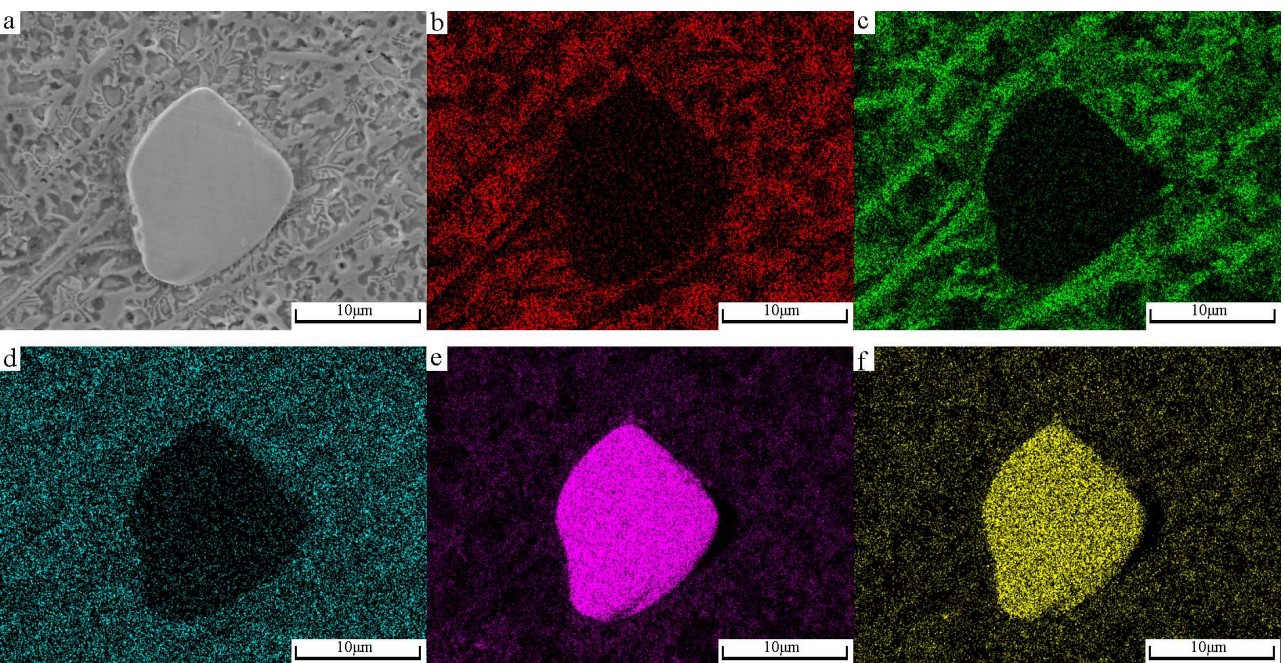

**Figure 8.** SEM micrograph and EDS mapping of the particle site of cladding coating with $Y_2O_3$ mass fraction of 0.01: (**a**) SEM micrograph; (**b**) Ni; (**c**) Cr; (**d**) Fe; (**e**) W; (**f**) Y.

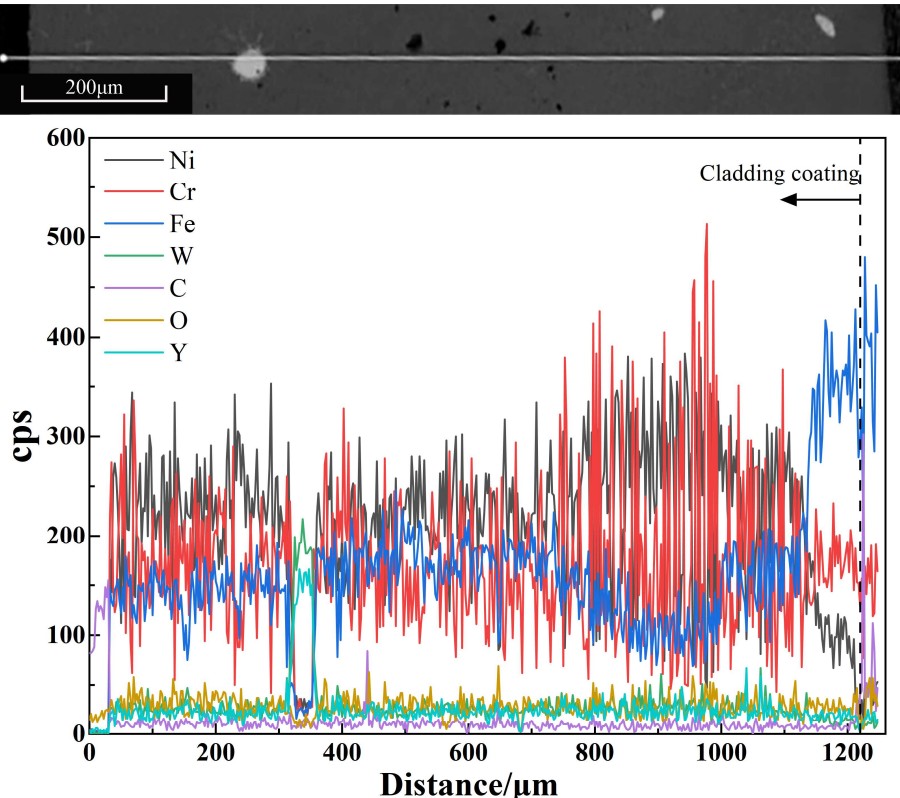

**Figure 9.** Element distribution of cladding coating with $Y_2O_3$ mass fraction of 0.01.

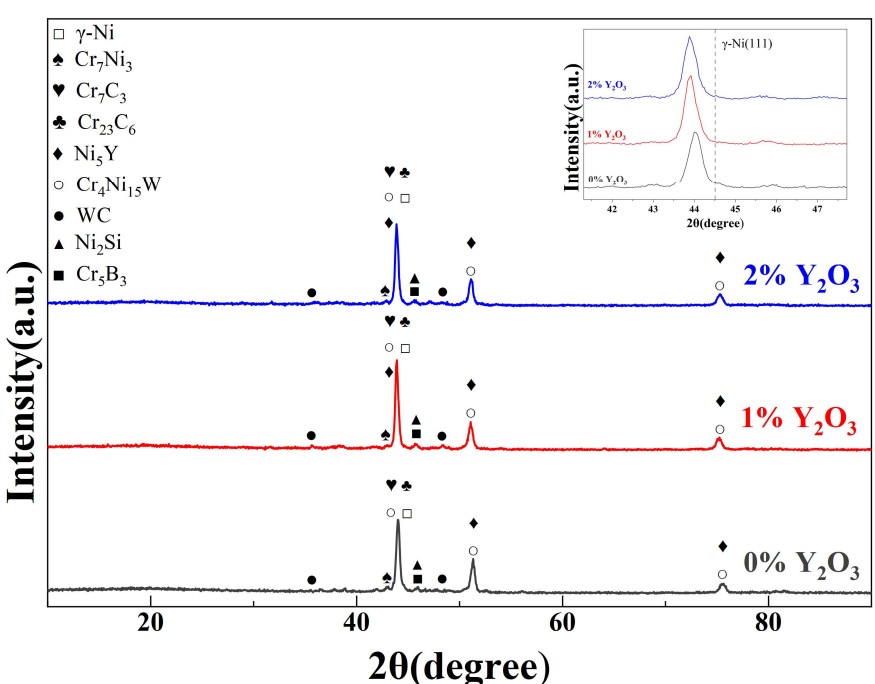

**Figure 10.** The XRD patterns of cladding coatings.

### 3.4. Microhardness Analysis of Cladding Coatings

Figure 11 is the microhardness distribution diagram of the samples, where the "HAZ" is the heat-affected zone. The average microhardness values of the cladding coatings and their standard deviations are shown in Figure 12. The microhardness distribution in the cladding coatings is even and the microhardness in the cladding coatings is dramatically higher than that in the substrates. The average microhardness of Ni-WC composite coatings reaches 655.7 HV. With the addition of $Y_2O_3$, the microhardness of the cladding coating increases somewhat. When the $Y_2O_3$ content reaches 1%, the microhardness increases to 712.4 HV. When the $Y_2O_3$ content increases to 2%, the microhardness decreases to 696.3 HV. As can be discerned from the analysis of microstructure and phase of the cladding coatings, the increase in microhardness is mainly due to the solid-solution-strengthening effect of the Cr and Fe entering $\gamma$-Ni in a solid solution manner [31], the dispersion strengthening effect of such hard phases as $Cr_{23}C_6$ and $Cr_7C_3$ formed by the WC in cladding coating and the C and Cr [32], and the refining of grain microstructure [33]. The appropriate amount of $Y_2O_3$ added enhances the solid-solution-strengthening effect and inhibits the growth of grain microstructure, the caused lattice distortion and grain refinement increase the resistance against dislocation motion, improving the strength of the cladding coating, and the improvement of fluidity of molten pool promotes the even distribution of hard phase, further increasing the microhardness of the cladding coating. The addition of $Y_2O_3$ at 2% reduces the fluidity of the melt pool, leading to an increase in grain size, which reduces the hardness. The continued addition of $Y_2O_3$ may further reduce the hardness, and may lead to defects such as cracks and porosity [25].

To further investigate the electrochemical corrosion property of coatings, Nyquist curves and equivalent circuits were tested and plotted for the samples, as shown in Figure 14, where R1 is the resistance of solution, C1 and R2 represent the capacitance and resistance between the surface passive film of cladding coating and the electrolyte solution, respectively, and C2 and R3 represent the capacitance and resistance between the working electrode and the electrolyte solution when the surface passive film is damaged, respectively [39]. In the Nyquist curves, Z' and Z" are the real and imaginary components of impedance, respectively. For a coating fabricated by laser cladding, the greater the impedance radius at low frequency, the better the corrosion resistance of the coating [40]. It can be seen that after $Y_2O_3$ is added, the capacitive reactance arc radius increases sig-

nificantly. When the $Y_2O_3$ mass fraction is 0.01, the radius at low frequency is the largest, which is consistent with the corrosion property reflected by $I_{corr}$ in Table 5. In summary, when 1% of $Y_2O_3$ is added in Ni60/WC cladding coating, it can most effectively improve the corrosion resistance property of the cladding coating in seawater. The electrochemical corrosion of Ni-based WC alloy coating occurs between carbide particles and bonding agent in the whole body of the binding agent, and at such defects in cladding coating as microcracks, and the most important corrosion mechanism is galvanic corrosion between the carbide particles and the surrounding bonding agent [38]. Thus, the $Y_2O_3$ added reduces the quantity of inclusions in the cladding coating, having a microstructure purifying effect, promotes the evenness of various components in the cladding coating, decreasing the size of carbide particles, and causes the grain microstructure to become even more dense, clogging the corrosion channels [41]. Excess $Y_2O_3$ added causes the microstructure of the cladding coating to become large and uneven, and large quantities of inclusions increase the quantity of primary cells, thus degrading the corrosion resistance property of the cladding coating.

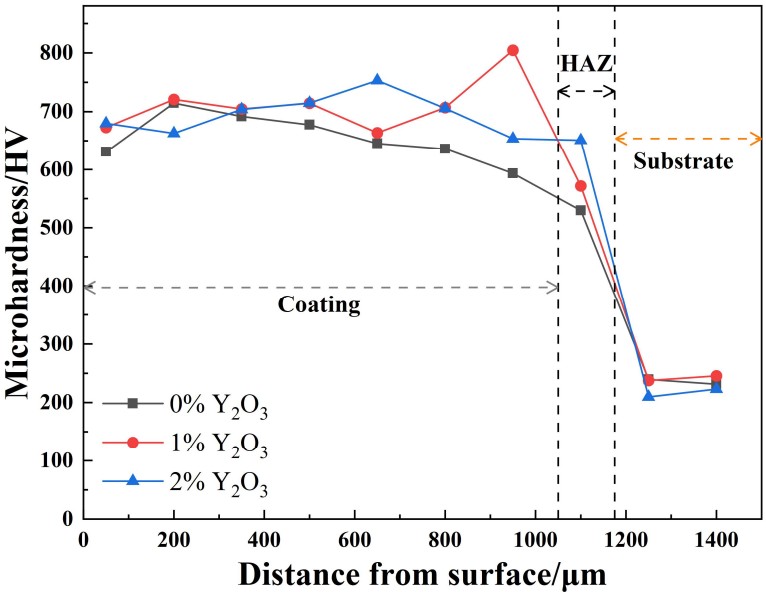

**Figure 11.** Microhardness distributions of the cladding coatings.

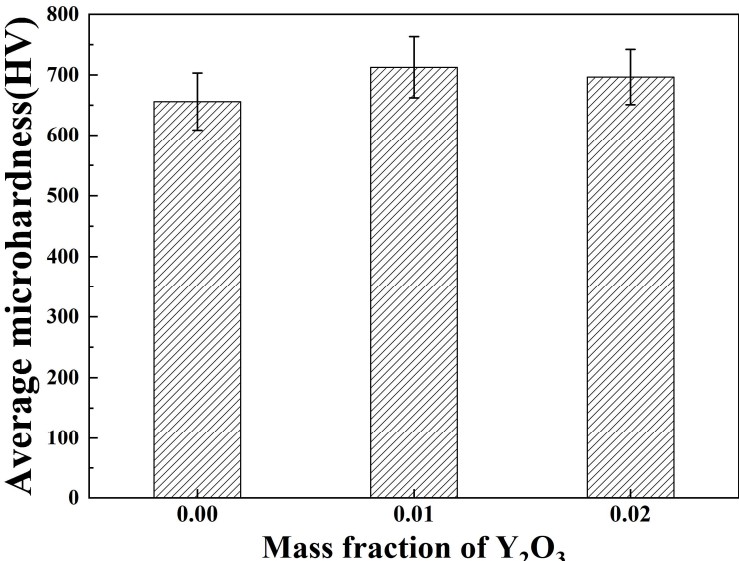

**Figure 12.** Average microhardness diagram of the cladding coatings.

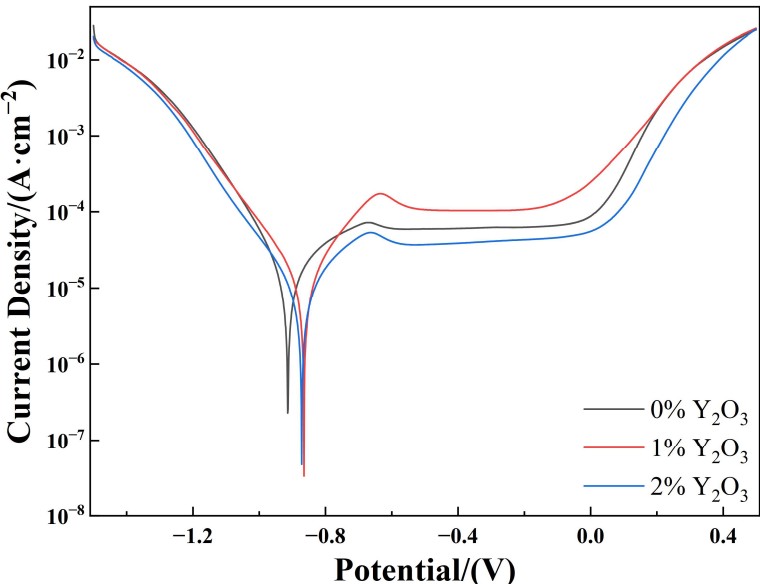

**Figure 13.** Potentiodynamic polarization curves of cladding coatings in 3.5 wt% NaCl solution.

**Table 5.** Electrochemical parameters of cladding coatings in 3.5 wt% NaCl solution.

| Mass Fraction of $Y_2O_3$ | $E_{corr}$/(V) | $I_{corr}$/($\mu A \cdot cm^{-2}$) |
| --- | --- | --- |
| 0 | −0.914 | 0.2292 |
| 0.01 | −0.865 | 0.0347 |
| 0.02 | −0.872 | 0.0476 |

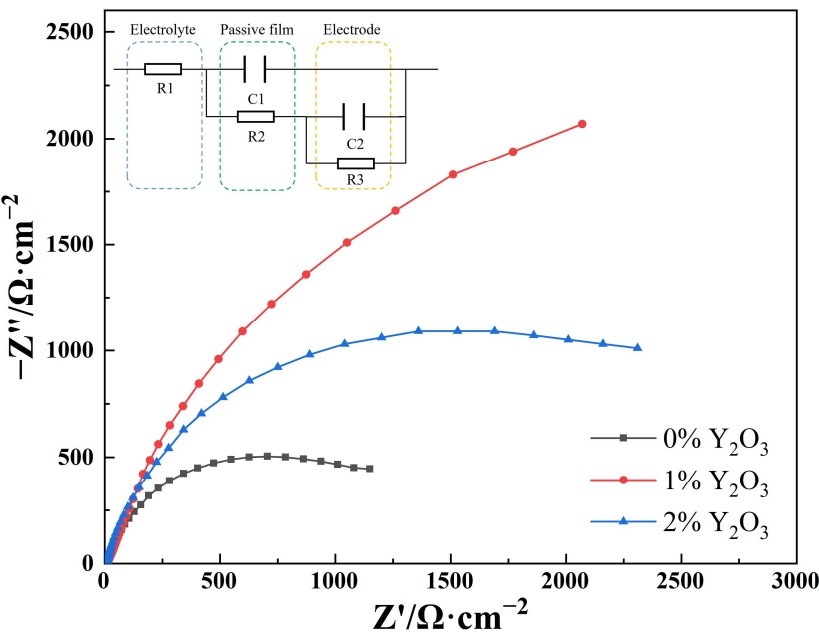

**Figure 14.** Equivalent circuit diagram and Nyquist curves of cladding coatings in 3.5 wt% NaCl solution.

### 3.5. Wear Resistance Analysis of Cladding Coatings

Figure 15 shows the frictional coefficients of modified composite coatings with different $Y_2O_3$ contents and Table 6 shows the wear loss values of the coatings. After 1–2 min, the frictional coefficient curve of a cladding coating gradually tends to be stable and exhibits a slow ascending trend. The frictional coefficient of a cladding coating decreases with an

increase in $Y_2O_3$ content; however, the frictional coefficient of the cladding coating with 2% $Y_2O_3$ added exhibits a more evident ascending trend over time. The reason is that a little $Y_2O_3$ added can effectively refine the reinforcing phase, ensuring even distribution of dispersant in the cladding coating, thus ensuring that the cladding coating has very high resistance to plowing by the microbumps on the surface of GCr15 balls. The dendritic carbide evenly distributed in the cladding coating promotes the transfer of load to the tough substrate phase and releases stress through coordinated deformation, thus avoiding brittle fracture of the carbide. In addition, grain refinement and the solid-solution-strengthening effect enhance the strength and microhardness of the cladding coating, thus improving the wear resistance property of the cladding coating [42]. Under a long time of wear, the carbide-reinforcing phase will come off gradually, thus resulting in degradation of the wear resistance and an increase in the frictional coefficient. The microstructure of the cladding coating with 2% $Y_2O_3$ added is distributed evenly, with a relatively large portion of the carbide-reinforcing phase distributed, so the wear resistance property is better in the initial stage of wear; however, when the $Y_2O_3$ addition amount reaches 2%, the bonding strength between the carbide-reinforcing phase and the Ni substrate may be reduced, so the frictional coefficient of the cladding coating with 2% $Y_2O_3$ added has the largest increase amplitude over wear time. If the content of $Y_2O_3$ in the cladding coating continues to increase, the hardness of the coating decreases rapidly. Excessive $Y_2O_3$ forms refractory compounds that affect the internal organization of the cladding coating, leading to weakening of the wear resistance [26].

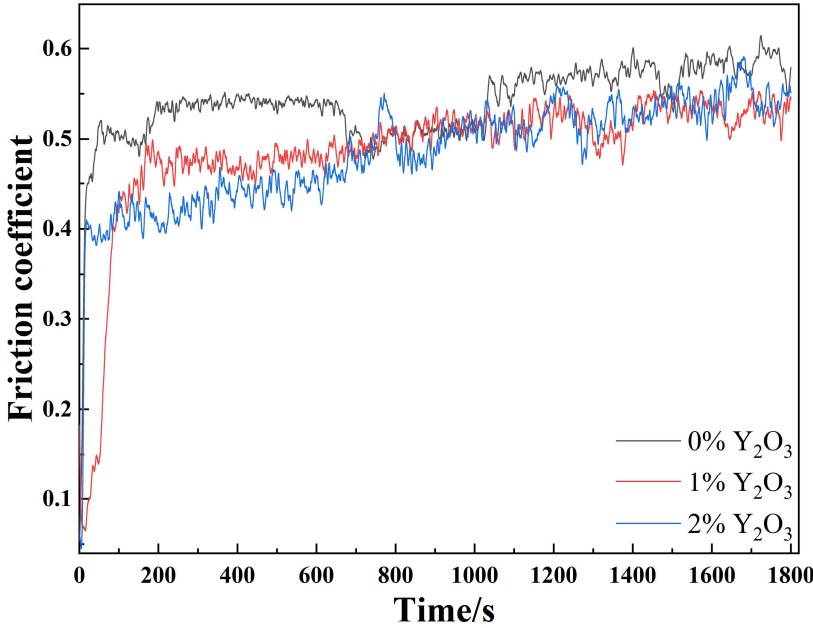

**Figure 15.** Frictional coefficients of the cladding coatings with different $Y_2O_3$ contents.

**Table 6.** Wear loss of cladding coatings with different $Y_2O_3$ contents.

| Mass Fraction of $Y_2O_3$ | Wear Loss/mg |
| --- | --- |
| 0 | 0.6 |
| 0.01 | 0.5 |
| 0.02 | 0.4 |

## 4. Conclusions

Ni-based/WC/$Y_2O_3$ coatings were fabricated on the surface of 316L stainless steel by laser cladding technology, and the following conclusions were drawn by analyzing the microstructure, microhardness, electrochemical corrosion property and wear resistance property of the coatings with different $Y_2O_3$ mass fractions:

(1) The fabricated composite coatings have good shaping, with only several air pores formed. It is discerned from observation of the cross-sections of the cladding coatings that all cladding coatings can achieve good metallurgical bonding with the corresponding substrate. Each cladding coating contains only a few incompletely molten WC particles. An appropriate amount of $Y_2O_3$ added inhibits the formation of the carbide-reinforcing phase, and the $Y_2O_3$ added causes the microstructure of the cladding coating to be more evenly distributed.

(2) The $Y_2O_3$ added has little effect on the phase composition of the cladding coatings. The rare earth element enters the $\gamma$-Ni in a solid solution manner, promoting the lattice distortion and increasing the interplanar spacing, and the rare earth element added bonds with Ni to form an $Ni_5Y$ phase in the cladding process.

(3) The average microhardness of Ni-based/WC coating reaches 655.7 HV. With the increase in $Y_2O_3$ content, the microhardness of the cladding coating increases to 712.4 HV first and then decreases to 696.3 HV, which, however, is over three times higher than that of the substrate.

(4) An appropriate amount of rare earth element can effectively improve the corrosion resistance property of the cladding coating in 3.5 wt% NaCl solution. When the $Y_2O_3$ mass fraction is 0.01, the cladding coating has the best corrosion resistance capability.

(5) The rare earth element added can effectively improve the wear resistance property of the cladding coating. With the increase in $Y_2O_3$ mass fraction, the wear loss decreases from 0.6 to 0.4.

**Author Contributions:** Conceptualization, F.L.; Data curation, F.L.; Funding acquisition, W.S.; Investigation, W.S. and Z.Z.; Methodology, F.L.; Project administration, W.S.; Resources, W.S.; Software, Z.Z.; Supervision, K.L.; Writing—original draft, F.L.; Writing—review & editing, F.L. All authors have read and agreed to the published version of the manuscript.

**Funding:** Supported by the Natural Science Foundation of China (62073089) and the Special Fund for Key Projects of Colleges and Universities in Guangdong Province(2020ZDZX2061).

**Institutional Review Board Statement:** Not applicable.

**Informed Consent Statement:** Not applicable.

**Data Availability Statement:** Not applicable.

**Conflicts of Interest:** The authors declare no conflict of interest.

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
