# Peer review of "Effect of Y2O3 Content on Microstructure and Corrosion Properties of Laser Cladding Ni-Based/WC Composite Coated on 316L Substrate"

_coatings, doi:10.3390/coatings13091532_

Round 1
Reviewer 1 Report
The authors studied the Effect of Y2O3 Content on Microstructure and Properties of Laser Cladding Ni-Based/WC Composite Coating. The work is prospective and includes many useful results.
The authors should add some more information in introduction regarding the goal of adding Y2O3 to the coating. What is the mechanism of it’s influence on the improvement of material properties?
The authors should outline the most appropriate coatings for improoving corrosion resistance and other properties.
Can the authors explain why they did not calculate the surface energy of the tested coatings? This is an important value in the hydrophilicity study of coatings.
Can the authors explain why they applied only one load 200 g for the microhardness test?
Minor editing of English language required
Reviewer 2 Report
Comments:
0. Major revision. 1. The novelty of this study should be inserted in the text clearly. 2. The advantages and disadvantages of this study should be investigated. 3. How does your paper contribute to the advancement of knowledge? 4. What are the gap areas and the new contribution in the paper? 5. The conclusion should be improved with clear quantitative findings. 6. The manuscript is satisfactory, however, a careful check is needed. 7. The “introduction” and “results and discussion” sections of the manuscript can be strengthened and supported with some papers related to the literature and cited.
Comments:
0. Major revision. 1. The novelty of this study should be inserted in the text clearly. 2. The advantages and disadvantages of this study should be investigated. 3. How does your paper contribute to the advancement of knowledge? 4. What are the gap areas and the new contribution in the paper? 5. The conclusion should be improved with clear quantitative findings. 6. The manuscript is satisfactory, however, a careful check is needed. 7. The “introduction” and “results and discussion” sections of the manuscript can be strengthened and supported with some papers related to the literature and cited.
Reviewer 3 Report
The researchers aimed to study the enhancement of corrosion resistance and durability of 316L stainless steel in marine settings. They employed laser cladding to create Ni-based/WC/Y2O3 layers with varying Y2O3 levels on the steel. Y2O3 addition refined the structure, increased hardness, wear resistance, and corrosion resistance in a salt solution. The best results were at 1% Y2O3 content. Overall, coatings with Y2O3 performed better.
For the abstract section, “…using a variety of characterization methods”, requires proper elaboration to highlight all the specific tools involved.
The introduction section is quite lengthy. Suggest to revise accordingly.
For the contents,
- SEM image as in Figure 1 should be put in the latter section “Results and discussion”, as it already considered as part of the results
- Rename “2.2 Subsection” for a more appropriate title
- Conclusion section need to revise. No need to do numbering, as normal paragraph should summarize well overall findings.
- Overall, most figure caption is too short, suggested to elaborate further
English language require certain degree of revision, especially at certain part of the manuscript. Kindly revisit and revise accordingly.
Reviewer 4 Report
Under the title ‘Effect of Y2O3 Content on Microstructure and Properties of Laser Cladding Ni-Based/WC Composite Coating’ the authors have figured out useful results for the marine engineering applications. In my point of view, the outcomes are interesting, properly arranged and may be accepted after the moderate revision and fulfillment of other formal requirements of MDPI-Coatings. But before publishing in this journal following remarks need to be addressed.
1. In Fig. 9, label and units on X- and Y- axes are missing.
2. Page 10, line 244, please rephrase the ‘The 2θ angle shifts to relatively low direction’ smething like ‘peak shitig to the low 2θ angles’.
3. In Fig.10, the XRD pattern showing about 10 different crsytalline phases. It is surprising why there are few diffraction peaks? Authors should explain the reason.
4. In the caption of Figure 10, correct pattern to patterns.
5. In the section 3.3, page 10 as ‘When the Y2O3 content increases to 2%, the microhardness decreases to 696.3 HV’ why the microhardness decreases for Y2O3 content more than 1%? What would happen if Y2O3 content increases even more than 2%? Please discuss.
6. In connetion with previous remarks (5), what would authors suggest about the correlation between the Y2O3-contents, microhardness and wear loss. Please also discuss it in the section 3.2. Wear resistance.
7. With the help of the literature, provide a comparison table containing the wear loss and other parameters, as obtained in this study, of few materials used for marine engineering purposes.
8. In Fig. 11, A region is mentioned by ‘HAZ’. Please mention its meaning.
9. In various figs, better to write units, on axes, all in the same style.
10. The title may be improved as ‘Effect of Y2O3 Content on Microstructure and Corrosion Properties of Laser Cladding Ni-Based/WC Composite Coated on 316L Substrate’.
no
Round 2
Reviewer 2 Report
Accept
Accept
